# Scaffold nucleoporins Nup188 and Nup192 share structural and functional properties with nuclear transport receptors

Kasper R Andersen[1†], Evgeny Onischenko[2†], Jeffrey H Tang[3], Pravin Kumar[1], James Z Chen[1], Alexander Ulrich[1], Jan T Liphardt[3], Karsten Weis[2*], Thomas U Schwartz[1*]

[1]Department of Biology, Massachusetts Institute of Technology, Cambridge, United States; [2]Department of Molecular and Cell Biology, University of California, Berkeley, Berkeley, United States; [3]Department of Physics, University of California, Berkeley, Berkeley, United States

**Abstract** Nucleocytoplasmic transport is mediated by nuclear pore complexes (NPCs) embedded in the nuclear envelope. About 30 different proteins (nucleoporins, nups) arrange around a central eightfold rotational axis to build the modular NPC. Nup188 and Nup192 are related and evolutionary conserved, large nucleoporins that are part of the NPC scaffold. Here we determine the structure of Nup188. The protein folds into an extended stack of helices where an N-terminal 130 kDa segment forms an intricate closed ring, while the C-terminal region is a more regular, superhelical structure. Overall, the structure has distant similarity with flexible S-shaped nuclear transport receptors (NTRs). Intriguingly, like NTRs, both Nup188 and Nup192 specifically bind FG-repeats and are able to translocate through NPCs by facilitated diffusion. This blurs the existing dogma of a clear distinction between stationary nups and soluble NTRs and suggests an evolutionary relationship between the NPC and the soluble nuclear transport machinery.

*For correspondence:
kweis@berkeley.edu (KW);
tus@mit.edu (TUS)

†These authors contributed equally to this work

## Introduction

Nucleocytoplasmic transport across the nuclear envelope is controlled and facilitated by nuclear pore complexes (NPCs), very large, modular protein assemblies (*Brohawn et al., 2009*; *Capelson et al., 2010*; *Aitchison and Rout, 2012*). NPCs are positioned in circular openings within the nuclear envelope, generated by the fusion of inner and outer nuclear membrane. About 30 different proteins, collectively called nucleoporins or nups, make up an NPC. These nups are arranged around a central eightfold rotational symmetry axis along the central pore opening. Cryo-electronmicroscopic (EM) studies have shown that the principal architecture of the NPC is conserved among all eukaryotes, however metazoan NPCs are more elaborate and have additional features compared to single cell organisms, explaining in part substantially different size estimates for the assembly (60–120 MDa vs 40–50 MDa) (*Reichelt et al., 1990*; *Alber et al., 2007*; *Brohawn et al., 2009*; *Grossman et al., 2012*). The entire structure is built around a central, donut-shaped scaffold, with prominent extensions that distinguish both the cytoplasmic and the nucleoplasmic faces of the NPC. The center of the NPC is thought to be filled by unstructured, highly repetitive protein extensions characterized by phenylalanine-glycine dipeptides (FG repeats), which are found in ~1/3 of all nups (*Rout et al., 2000*; *Cronshaw et al., 2002*). FG repeats are critical for NPC function and allow for selective passage of transport cargos that are bound to soluble nuclear transport receptors (NTRs) (*Görlich and Kutay, 1999*; *Weis, 2003*). The mechanistic details of how FG repeats contribute to NPC selectivity remain somewhat controversial, however, it has been firmly established that all soluble NTRs specifically bind to FG-repeat

**eLife digest** The nucleus of a cell is surrounded by a two-layered membrane that controls the flow of molecules from the cytoplasm into the nucleus and vice versa. The molecular traffic between the cytoplasm and nucleus is essentially controlled by nuclear pore complexes—large, multi-protein structures that are embedded in the membrane. Each nuclear pore complex contains about 30 different proteins called nucleoporins or nups, which combine to form a structure with a central pore that allows the molecules to enter and leave the nucleus.

The centre of the nuclear pore complex is thought to be filled with protein filaments that contain a large number of so-called FG repeats (where F and G are the amino acids phenylalanine and glycine). Specialized molecules called soluble nuclear transport receptors, which carry various cargoes between the cytoplasm and nucleus, can bind to these FG repeats, and the interaction between the receptors and the FG repeats is crucial for the selective transport of molecules between the cytoplasm and the nucleus.

The large size of the nuclear pore complex has hindered efforts to work out its structure, but in recent years researchers have been able to obtain structures for many individual nups and their subcomplexes. Now, Andersen et al. have determined the structure of one of the largest nups, Nup188. This has led to the discovery that it and a related nup, Nup192, share unexpected features with soluble nuclear transport receptors.

In general the first step when attempting to determine the structure of a biomolecule is to form a crystal. Since full-length Nup188 did not crystallize, Andersen et al. instead crystallized two large fragments of Nup188, determined the structures of these fragments, and then combined these to produce the likely structure of the full-length protein. They found that Nup188 has a structure that consists of stacked helices and is more flexible than other nups. Moreover, its structure was very similar to those of soluble nuclear transport receptors, and this led Andersen et al. to investigate whether Nup188 also had similar functional features.

Surprisingly, they discovered that both Nup188 and Nup192 could bind FG repeats, just like nuclear transport receptors. What is more, this binding allowed both nups to travel through nuclear pore complexes in *in vitro* transport reactions. These findings have implications for the understanding of the organization and function of FG-repeats and suggest that the stationary elements of the nuclear pore complex and soluble nuclear transport receptors are evolutionarily related.

domains and that interactions between NTRs and FG repeats are critical for nucleocytoplasmic transport. The central scaffold of the NPC donut structure is formed by approximately 15–20 nups that are organized into four major subcomplexes. These are the Y- (or Nup84) complex, the Ndc1 complex, the Nsp1 complex, and the Nic96 complex. The Y-complex is essential for NPC biogenesis and has been well characterized. It contains seven universally conserved proteins that form a highly branched, Y-shaped structure (*Lutzmann et al., 2002*; *Kampmann and Blobel, 2009*). Currently ~90% of the complex has been elucidated in crystallographic detail (*Berke et al., 2004*; *Brohawn et al., 2008*; *Brohawn and Schwartz, 2009*; *Leksa et al., 2009*; *Whittle and Schwartz, 2009*). From this analysis it is evident that the structural elements, and also their interactions are shared with vesicle coats, specifically the outer coat of COPII vesicles (*Devos et al., 2004*; *Brohawn et al., 2008*; *Onischenko and Weis, 2011*). The Ndc1 complex contains proteins thought to anchor the NPC in the highly-curved pore membrane (*Onischenko et al., 2009*). The Nsp1 complex is built from three coiled-coil proteins, each with prominent FG-repeats. The Nsp1 complex is centrally located in the NPC, with the FG-repeats contributing to the selective transport barrier (*Hülsmann et al., 2012*), and the coiled-coil moiety functioning as a bridge to the other NPC scaffold units (*Grandi et al., 1995*; *Bailer et al., 2001*).

The Nic96 complex organizes four architectural proteins, Nic96, Nup157/170, Nup188, and Nup192, as well as the more disordered protein Nup53/59 (*Nehrbass et al., 1996*; *Marelli et al., 1998*; *Theerthagiri et al., 2010*; *Amlacher et al., 2011*). Structures for large fragments of Nic96 and Nup170 are available and show distinct stacked helical arrangements (*Jeudy and Schwartz, 2007*; *Schrader et al., 2008*; *Whittle and Schwartz, 2009*).

Nic96 interacts directly with the Nsp1 complex (*Bailer et al., 2001*; *Schrader et al., 2008*), while Nup53/59 binds directly to the pore membrane and interacts with other Ndc1 complex components

(*Onischenko et al., 2009*). This suggests that the Nic96 complex functions to bridge the pore membrane to the central Nsp1 complex. Intriguingly, Nic96 and its vertebrate ortholog Nup93 were shown to bind FG-repeats in vitro (*Schrader et al., 2008*; *Xu and Powers, 2013*).

Nup188 and Nup192 are the last two large nucleoporins whose atomic structures are unknown. To obtain structural information for these two related proteins, we set out to crystallize Nup188. We succeeded in solving the structures of two fragments, together covering 85% of the 202 kDa full-length protein. Nup188 is a stacked helical protein with intrinsic flexibility. Interestingly, a structural comparison reveals similarity between Nup188 and NTRs. Moreover, we can show that both Nup188 and Nup192 specifically bind to FG-repeats and translocate across NPCs by facilitated diffusion. These findings blur the line between the strict separation of soluble NTRs and stationary nups, question some of the current models of FG-repeat organization, and suggest that the NPC and the soluble transport machinery are evolutionary related.

## Results

### The structure of Nup188

To determine the structure of Nup188 we first expressed full-length protein from *Saccharomyces cerevisiae* (Sc) and the thermophilic fungus *Myceliophthora thermophila* (Mt) (*Berka et al., 2011*). Although the *Mt* homolog behaved better in vitro, it did not yield crystals of the full-length protein. Using a series of truncation constructs, generated through a combination of limited proteolysis, phylogenetic analysis and structure prediction, we were able to obtain crystals of two major parts of *Mt*Nup188. The N-terminal fragment (Nup188N) includes residues 1–1160 and its structure was solved to 2.65 Å resolution (*Figure 1* and *Table 1*). The C-terminal fragment (Nup188C) includes residues 1445–1827, and we solved its structure to 3.0 Å resolution (*Figure 2* and *Table 1*). Each structure was solved using single-anomalous dispersion (SAD) experimental phases derived from crystals grown from selenomethionine-labeled protein. Nup188N crystallized in space group $C2_1$ with one copy per asymmetric unit. With a Wilson B-factor of 45 Å$^2$ the structure is generally well defined. Nup188C crystallized in space group $P2_12_12_1$ with two copies per asymmetric unit. Compared to Nup188N, it is not quite as well defined (Wilson B-factor is 75 Å$^2$), likely because those crystals have a higher solvent content and are more loosely packed.

Nup188N is built from 52 stacked helices. They fold into a large right-handed superhelical ring that resembles a lock washer (*Figure 1B*). Although the overall structure has a general repeat pattern and is built around 2-helix (HEAT repeat), and 3-helix (ARM-repeat) elements, several distinct details specific to Nup188 make it a rather unique protein. We can discern an inner ring of helices forming the concave surface and an outer ring of helices forming the convex surface creating three distinct subdomains (*Figure 1A,B*). The first subdomain consists of 15 helices that are arranged in irregular orientations without an obvious repeat pattern. Next, helices α16–25 make up subdomain II and are organized around three HEAT repeats (α16/17, α18/19, α22-23). Helix α23 protrudes from the molecule and is sandwiched between helices α24 and α25, such that they are splayed apart at a 90° angle. In consequence, the helical stack pattern is interrupted. Subdomain III includes helices α26 to α52 and is structured around the second helical stack element, pivoted against subdomain II. It has 10 helical repeats of both the HEAT and the ARM type and forms two curved layers of helices. Between helix α32 and α33, we find a β-stranded insertion with structural homology to Src-homology (SH3) domains (see *Figure 1D*). The Nup188N ring structure is closed by four helices (α10–α13) from sub-domain I that latch onto a complementary surface created by helices α44, α46 and α47. The interface is mixed in character, with van der Waals contacts as well as polar and ionic interactions. In total, the latch covers an 1511 Å$^2$ interface (*Figure 1C*).

As noted, Nup188 contains an inserted β-stranded-domain, which has structural similarity to the SH3 domain superfamily (*Li, 2005*). The domain is arranged in five anti-parallel β-strands that fold into a β-barrel, characteristic for the SH3 protein family (*Figure 1D*). Sequence alignment and secondary structure prediction of Nup188 from a broadly diverged range of species indicate that all Nup188 orthologs contain this SH3-like domain and that the position within the protein is conserved (data not shown). In contrast to canonical SH3 domains, the SH3-like domain in Nup188 has a different β-strand topology, which represents a circular permutation (*Figure 1D*). SH3 domains are protein interaction domains known to predominantly bind proline-rich peptides (*Ren et al., 1993*), although non-consensus ligands have also been found in recent years (*Saksela and Permi, 2012*). The surface residues typically

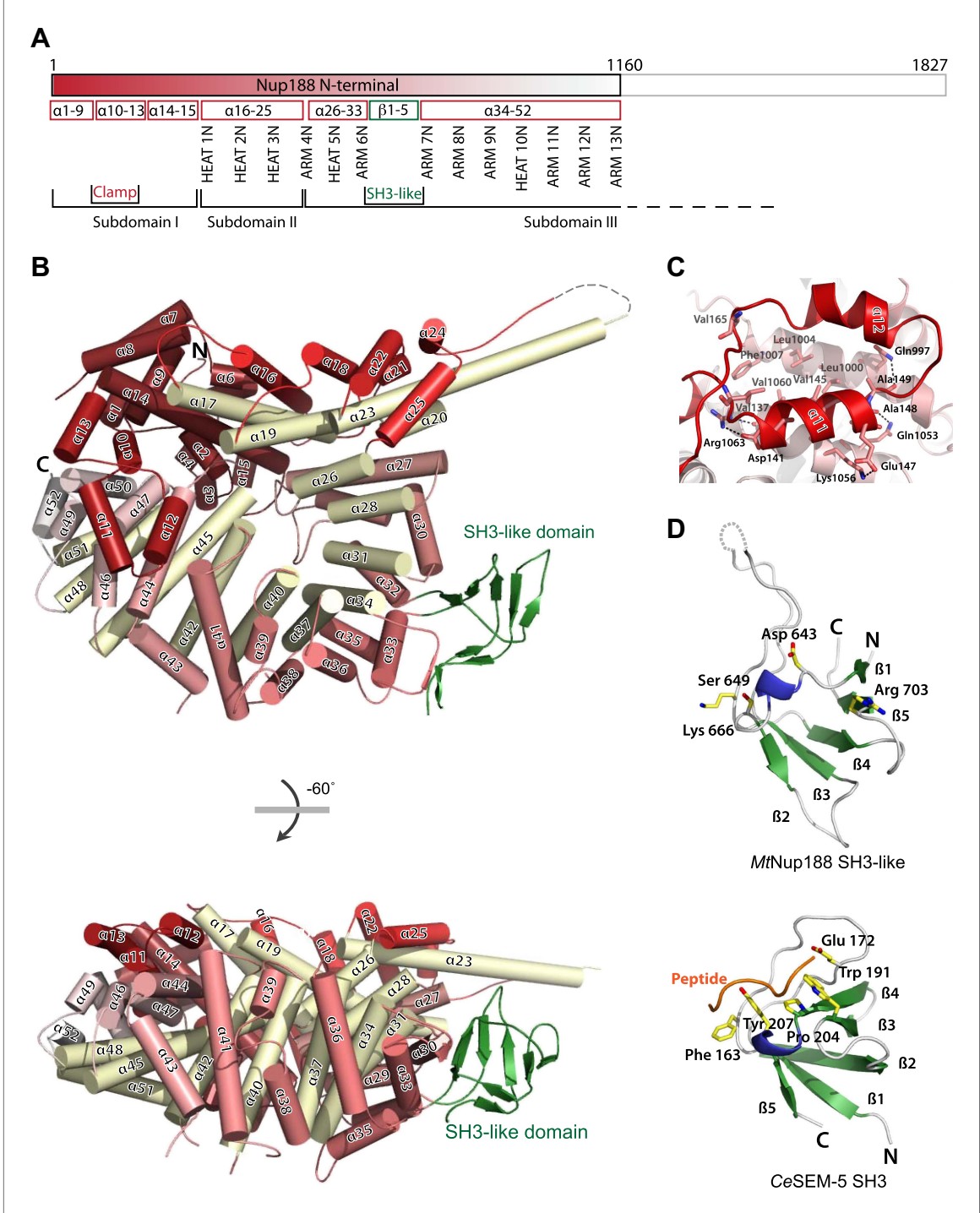

**Figure 1**. Crystal structures of Nup188N. (**A**) Schematic diagram of full-length *Myceliophthora thermophila* Nup188N with details about the subdomain arrangement. The crystallized fragment is boxed and gradient-colored in red with helical HEAT- and ARM-repeats indicated. (**B**) The crystal structure of Nup188N is shown in cartoon representation. The helices are gradient-colored as in (**A**). The inner helical ring is in pale-yellow to highlight the superhelical organization of the protein. The SH3-like domain insert is shown in green. (**C**) Close-up of the ring-closing latch. The clamping helices α11 and α12 (red) contact a substantial surface area formed by helices 44, 46, and 47. (**D**) The SH3-like domain of Nup188 compared to the canonical, peptide-bound SH3 domain of Sem-5 from *C. elegans* (PDB 1SEM). Conserved residues important for peptide interaction are labeled. Note that these residues are not conserved in the SH3-like domain of Nup188. In addition, the SH3-like domain is a circular permutation of the canonical SH3-domain.

**Table 1.** X-Ray data collection and refinement statistics

| | *Mt*Nup188N (1–1160) | *Mt*Nup188N (1–1160) | *Mt*Nup188C (1445–1827) |
|---|---|---|---|
| PDB code | 4KF7 | | 4KF8 |
| **Data collection** | | | |
| Data set | Native | Selenomethionine | Selenomethionine |
| Space group | C2 | C2 | $P2_12_12_1$ |
| a, b, c (Å) | 169.0, 94.8, 91.6 | 168.3, 94.6, 91.4 | 64.4, 66.7, 162.5 |
| α, β, γ (°) | 90, 98.9, 90 | 90, 98.4, 90 | 90, 90, 90 |
| Wavelength (Å) | 0.9792 | 0.9792 | 0.9792 |
| Resolution range (Å) | 66.8–2.65 (2.72–2.65) | 90.4–2.90 (3.02–2.90) | 81.1–3.00 (3.11–3.00) |
| Total reflections | 170,859 | 209,695 | 50,886 |
| Unique reflections | 41,673 | 58,249 | 14,539 |
| Completeness (%) | 100 (99.9) | 99.9 (98.8) | 99.9 (99.7) |
| Redundancy | 4.1 (3.8) | 3.6 (3.1) | 3.5 (3.1) |
| $R_{merge}$(%) | 3.1 (77.2) | 3.0 (32.8) | 2.7 (56.1) |
| $R_{p.i.m.}$(%) | 2.0 (51.0) | 4.1 (26.3) | 1.6 (27.0) |
| $I/\sigma(I)$ | 31.3 (1.8) | 9.6 (1.2) | 21.4 (4.8) |
| Wilson B factor (Å$^2$) | 51.6 | 50.0 | 75.0 |
| **Refinement** | | | |
| Resolution range (Å) | 66.8–2.65 | | 81.1–3.00 |
| $R_{work}$ (%) | 18.1 | | 27.5 |
| $R_{free}$ (%) | 22.5 | | 29.4 |
| **Number of reflections** | | | |
| Total | 41,661 | | 14,536 |
| $R_{free}$ reflections | 1919 | | 1139 |
| **Number of atoms** | | | |
| Protein | 8516 | | 4366 |
| Water | 128 | | 0 |
| B factors (Å$^2$) | | | |
| Protein | 64.4 | | 92.3 |
| Water | 44.9 | | |
| **RMSD** | | | |
| Bond length (Å) | 0.004 | | 0.003 |
| Bond angles (°) | 0.904 | | 0.769 |
| **Ramachandran plot** | | | |
| Favored (%) | 95.0 | | 92.2 |
| Allowed (%) | 4.1 | | 6.8 |
| Outliers (%) | 0.9 | | 1.0 |

The highest resolution shell is in parenthesis. $R_{merge}$ is the merging R factor. $R_{p.i.m.}$ is the precision-indicating merging R factor. For definitions, see **Weiss (2001)**.

involved in peptide binding in previously characterized SH3 domains are not present in Nup188. Instead Nup188 has a different set of residues that are conserved between Nup188 orthologs. This conservation suggests that Nup188 also binds a specific protein or peptide motif, but it most likely does not interact with known SH3 targets.

The C-terminal fragment of Nup188, Nup188C, is a right-handed arc-shaped superhelical structure built from 19 helices that form 6 helical repeats, which are stacked in regular order (**Figure 2**). The first helical pair (α1 and α2) forms a HEAT repeat followed by 5 ARM repeats. The inner face of the structure

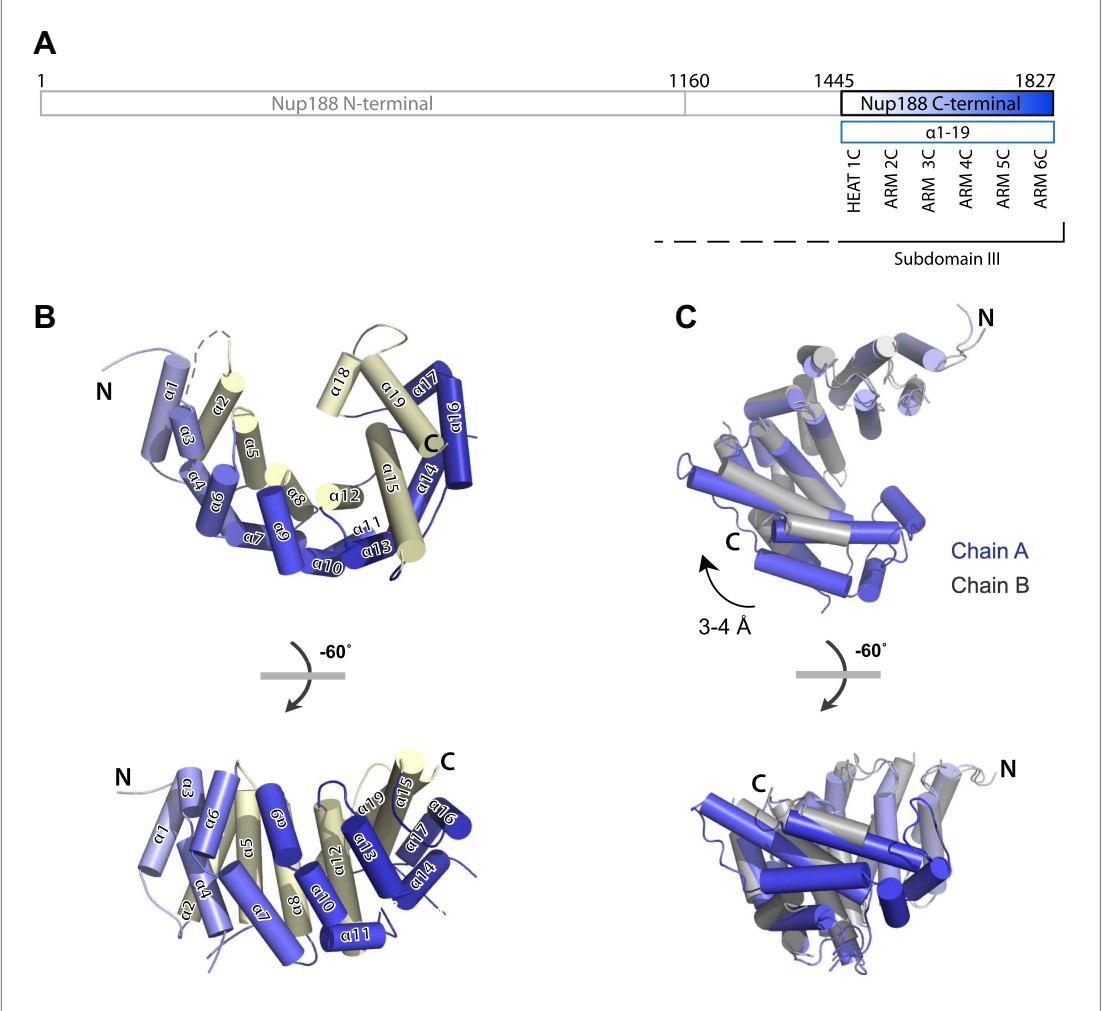

**Figure 2**. Crystal structures of Nup188C. (**A**) Schematic diagram of full-length *Myceliophthora thermophila* Nup188C with details about the subdomain arrangement. The crystallized fragments is boxed and gradient-colored in blue with helical HEAT- and ARM-repeats indicated. (**B**) Crystal structure of the C-terminal part of Nup188, gradient-colored as in (**A**), with the inner helices in pale-yellow. (**C**) Superposition of chains A (blue) and B (gray) of Nup188C shows the flexibility seen in the crystal, in which the outer helices of chain B move by 3–4 Å relative to the helices in chain A.

is built from helices α2, α5, α8, α12, α15, and α19. The C-terminal helix α19 caps the structure. At the N terminus, helices α1 and α2 expose a hydrophobic surface, which strongly suggests that the full-length protein continues in a stacked pattern. This hydrophobic surface engages in a twofold symmetric crystal packing contact with a neighboring Nup188C molecule resulting in a horseshoe-shaped dimer. Because this contact is generated by a protein truncation, we suggest that it is a crystal artifact and not physiologically relevant. Similar packing contacts are frequently observed with truncated, stacked helical proteins. Interestingly, the two Nup188C copies in the asymmetrical unit have a different curvature, likely reflecting the intrinsic flexibility of the molecule (*Figure 2C*).

Since we were unable to obtain crystals for full-length Nup188, we attempted to model the missing 283-residue segment (*Figure 3A*), taking various information into consideration. To start, secondary structure prediction suggests that the segment folds into 10 helices. Furthermore, the adjacent ends of both crystallized fragments, Nup188N and Nup188C, are regular repeats, therefore we reasoned that the missing segment very likely forms five helical pairs connecting the two parts into one continuous super-helical element (*Figure 3B*). This assumption is supported by 3D comparisons of the crystallized fragments with published structures of other stacked helical proteins. For both fragments we find superposable helical proteins in the PDB that extend in a repeat pattern further into the space where we expect the missing residues of Nup188 to be positioned. The assumption of a continuous helical structure is also

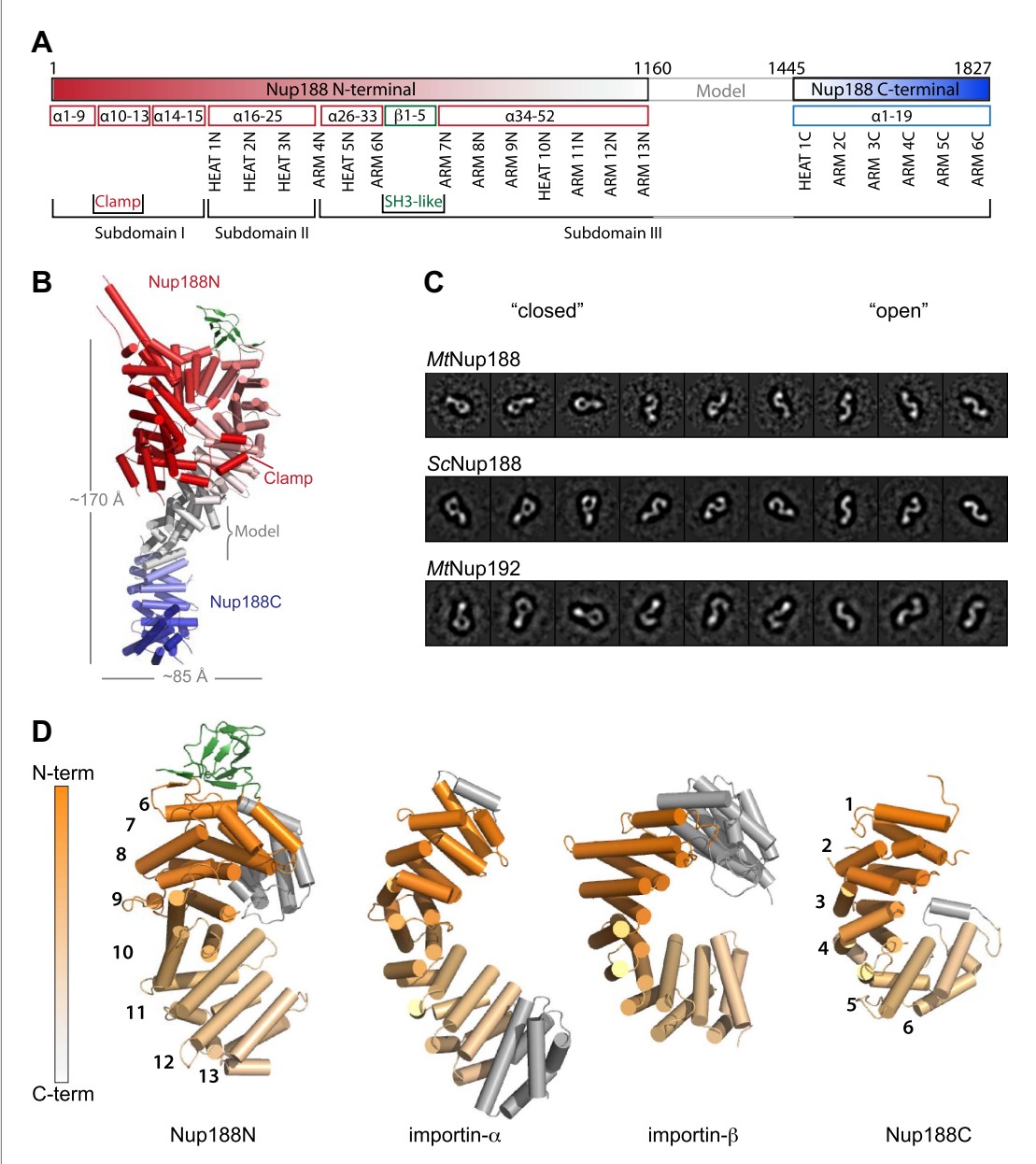

**Figure 3**. Structural details of Nup188 and comparison to other proteins. (**A**) Schematic diagram of full-length *Myceliophthora thermophila* Nup188 with details about the subdomain arrangement for the entire protein. (**B**) Full-length Nup188 gradient-colored in red-gray-blue. The modeled region and the overall dimensions are indicated. (**C**). Single-particle negative stain EM class averages of full-length *Mt*Nup188, *Sc*Nup188 and *Mt*Nup192. Classes are roughly grouped and are ordered from more 'closed' (left) to 'open' conformations (right). The three proteins are evidently structurally related. (**D**) Structural similarity between Nup188 and importin-α and -β. The aligned areas of Nup188N, Nup188C, importin-α (3L3Q) and importin-β (1QGR) are gradient-colored in orange from N to C terminus. Non-aligned helices are in gray and the repeat numbering is indicated.

supported by limited proteolysis experiments. Although *Mt*Nup188 is cut in numerous positions, the resulting fragments still behave like the full-length protein in a gel filtration experiment (data not shown).

To experimentally support our model of full-length *Mt*Nup188 we performed single particle EM analysis. We obtained 2D class averages from negatively stained particles (*Figure 3C*) demonstrating that *Mt*Nup188 can adopt multiple conformations. Several 2D classes are consistent with the crystal structures and show a ring-like structure with a bent, elongated extension. The overall dimensions are consistent with our full-length model measuring ~170 Å in length and ~85 Å across the N-terminal

ring. Other 2D classes show a twisted S-shaped molecule, where the ring is apparently opened. Thus, the protein can adopt various conformations in solution, consistent with the flexibility observed for many stacked helical domains.

## Nup188 and Nup192 are structurally related

Like many other architectural nucleoporins, Nup188 is rather poorly conserved in sequence across diverse eukaryotes (*Neumann et al., 2010*). To test whether Nup188 is conserved in structure, we analyzed Nup188 from *S. cerevisiae* by single-particle negative stain EM and compared it to the thermophilic ortholog. The two fungi are separated by ~800 million years in evolution (*Hedges et al., 2006*) and the Nup188 proteins share 15% sequence identity. As expected, *Mt*Nup188 and *Sc*Nup188 are very similar in structure, at least at the resolution obtained by EM analysis (*Figure 3C*). Both proteins show similar overall dimensions and share flexibility, evident in distinct 2D classes.

Nup188 and Nup192 are most likely paralogs that originate from an early gene duplication event (*Mans et al., 2004*). The two proteins are weakly conserved with an approximate sequence identity of 15%. The size range of 1600–2000 residues is also conserved among Nup188 and Nup192 (Nup205 in vertebrates). Binding data using the *Xenopus laevis* homologs Nup188 and Nup205 suggest that both proteins share some similar functions in the NPC as both compete for the same binding site on the nucleoporin Nup93 (Nic96 in yeast) (*Theerthagiri et al., 2010*). To analyze the structural similarity between Nup188 and Nup192, we next performed single particle negative stain EM of *Mt*Nup192 (*Figure 3C*). We found that the overall morphology of *Mt*Nup192 is very similar to Nup188, again with a ring structure of ~90 Å in diameter and a length of ~185 Å. Our data is consistent with the recent EM analysis of Nup192 from *Chaetomium thermophilum* (*Amlacher et al., 2011*) and like Nup188, Nup192 is also a flexible molecule in solution, with coexisting 'open' and 'closed' conformations.

Sequence analysis and comparison between Nup188 and Nup192 do not reveal a SH3-like domain insert in Nup192. No sequence conservation in this area can be observed between Nup188 and Nup192 and secondary structure predictions do not predict the presence of β-sheets in Nup192. We therefore conclude that the SH3-like domain is a specific feature of Nup188 that cannot be detected in Nup192.

## Structural similarity between Nup188 and nuclear transport receptors

Stacked helical proteins are abundantly found in the eukaryotic proteome, serving multiple biological functions (*Aravind et al., 2006*). To identify structures more closely related to Nup188 we used 3D similarity searches using our experimental structures as templates. The most regular part of Nup188N is subdomain III, which was found to be structurally similar to nuclear transport receptors (NTRs) (*Figure 3D*). A shared core of helical repeats superimposes with an RMSD of 3.6 Å to importin-α (276 C-α positions) and 4.5 Å to importin-β (263 C-α positions), respectively. Similarly we found that Nup188C is structurally similar to NTRs and superimposes with an RMSD of 3.2 Å to importin-α (208 C-α positions) and 3.7 Å to importin-β (212 C-α positions), respectively. Both Nup188 and NTRs form right-handed superhelices with different degrees of curvature. The RMSD values show that NTRs and Nup188 are generally related, as has been speculated previously based on EM- and 3D prediction methods (*Flemming et al., 2012*).

## Nup188 and Nup192 specifically bind FG-repeats of nucleoporins but do not interact with RanGTP

The intriguing structural similarity between Nup188, Nup192 and NTRs prompted us to test whether they also share other properties. We first set out to examine whether Nup188 and Nup192 share the unifying feature of all NTRs and interact with nucleoporin FG-repeats. *Mt*Nup188, *Mt*Nup192, importin β from *S. cerevisiae* (*Sc*Kap95) or 3xGFP were incubated with a selection of immobilized budding yeast FG-repeat domains and the interactions were monitored in pulldown experiments. To control for binding specificity all proteins were pre-mixed with excess bacterial extract proteins (*Figure 4—figure supplement 1A*). The panel of FG-repeat regions included Nup116(348–458), Nup100(1–310) and Nup100(1–610) representing GLFG-type repeats, and Nsp1(1–603) representing FXFG-type repeats (*Yamada et al., 2010*). In addition, we included the Nup116(348–458) 10xFA mutant, which lacks the key phenylalanine residues necessary for binding to NTRs (*Bayliss et al., 2000, 2002*; *Patel et al., 2007*). As expected, *Sc*Kap95 was firmly enriched (10-fold to 40-fold over bulk bacterial extract proteins) on both types of wild-type FG-repeat regions. This interaction was highly specific since *Sc*Kap95 did not bind to the Nup116(348–458)10xFA mutant and no interaction between FG-coated beads and 3xGFP could be detected (*Figure 4A,B*). Of note, *Sc*Kap95 was efficiently eluted with 1M

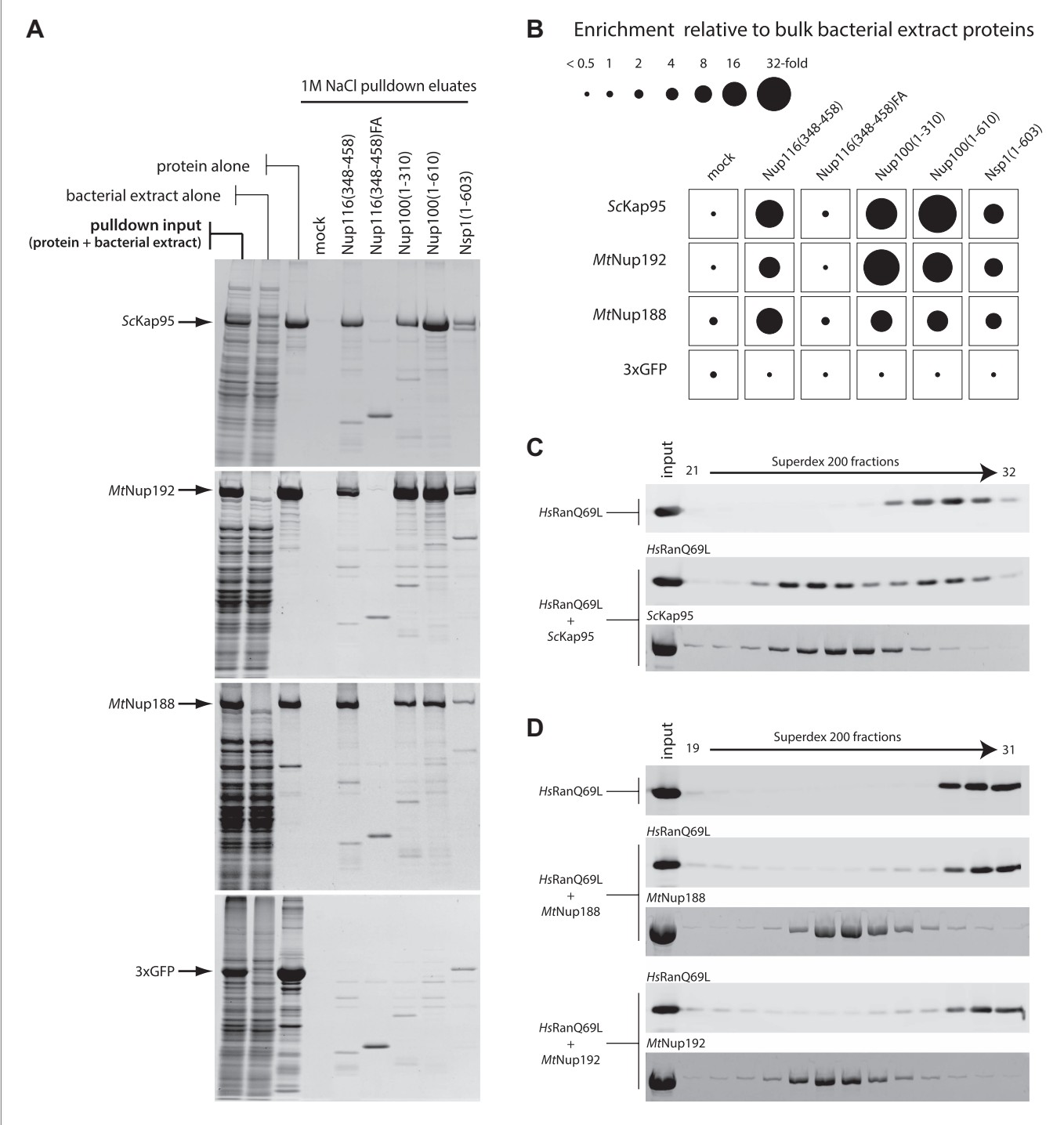

**Figure 4**. Biochemical properties of Nup188 and Nup192. (**A** and **B**) Binding of *Mt*Nup188 and *Mt*Nup192 to various FG-coated beads monitored by in vitro pulldown assays. (**A**) Standard volumes of the soluble protein mixtures (pulldown input), separate components of the mixtures (protein alone, bacterial extract alone) and the 1M NaCl pulldown eluates form various FG-coated beads (1M NaCl pulldown elates) were precipitated by methanol-chloroform and re-solubilized in a standard volume of SDS-sample buffer. The samples were separated by SDS-PAGE and the gels were stained with SYPRO-Ruby. (**B**) The diagram representing relative enrichments of *Sc*Kap95, *Mt*Nup188, *Mt*Nup192 and 3xGFP in various FG-pulldown eluates, as compared to bulk bacterial extract proteins. The enrichment values (ratios between the eluted and input protein amounts) were computed using protein band intensities from the SYPRO Ruby stained gels shown in (**A**). (**C**) *Hs*RanQ69L alone or an equimolar mixture of *Hs*RanQ69L with *Sc*Kap95 were subjected to gel filtration using a Superdex 200 column, and the eluted fractions were separated by SDS-PAGE. Ran was visualized by Western blotting

*Figure 4. Continued on next page*

Figure 4. Continued

against the ZZ-tag, and *Sc*Kap95 was visualized by SYPRO Ruby staining. (**D**) Superdex 200 elution profiles of ZZ-*Hs*RanQ69L (ZZ-tag Western blot) with or without the addition of *Mt*Nup188 or *Mt*Nup192 (SYPRO Ruby) analyzed as described in (**C**).
The following figure supplements are available for figure 4:

**Figure supplement 1**. Characterization of the FG-repeat interactions in pulldown assays.

NaCl from all the FG-coated beads consistent with the relatively weak interaction between *Sc*Kap95 and FG-repeat regions (***Pyhtila and Rexach, 2003***) (***Figure 4A***, ***Figure 4—figure supplement 1B***). Intriguingly, both *Mt*Nup188 and *Mt*Nup192 were strongly enriched on all wild-type FG-repeat regions and did not show significant binding to the Nup116(348–458)10xFA variant. We therefore conclude that both these nucleoporins can specifically bind to the FG-repeats in vitro and behave like the NTR *Sc*Kap95 in this assay (***Figure 4A,B***). The only notable deviation from the FG-binding pattern was observed with *Mt*Nup188, which was relatively inefficiently eluted with 1M NaCl from Nup100(1–310) and Nup100(1–610), suggesting that *Mt*Nup188 either interacts with these FG-repeat regions more tightly than *Sc*Kap95 or that these interactions are more hydrophobic in nature (***Figure 4—figure supplement 1B***).

NTRs of the importin-β/karyopherin family can also form direct complexes with the small GTPase Ran in its GTP-bound form (***Floer and Blobel, 1996***; ***Chook and Blobel, 1999***; ***Cook et al., 2007***). Sequence analysis did not detect any obvious RanGTP binding surface in either Nup188 or Nup192. Consistent with this, we were unable to detect binding between RanGTP and either *Mt*Nup188 or *Mt*Nup192 by gel filtration analysis (***Figure 4D***) whereas the NTR *Sc*Kap95 bound robustly to RanGTP (***Figure 4C***). We conclude that in contrast to NTRs neither *Mt*Nup188 nor *Mt*Nup192 can form a stable complex with RanGTP in our binding assay conditions.

## Nup188 and Nup192 translocate through intact NPCs by facilitated diffusion

A defining functional property of NTRs is their ability to mediate transport across the NPC. The structural similarity between *Mt*Nup188, *Mt*Nup192 and NTRs and their common ability to bind FG-repeats led us to ask whether *Mt*Nup188 and *Mt*Nup192 could also efficiently translocate through intact NPCs. To this end *Mt*Nup188 and *Mt*Nup192 were tagged with YFP, and their nuclear translocation was monitored by quantitative fluorescence microscopy using the standard in vitro nuclear transport assay in digitonin-permeabilized HeLa cells (***Adam et al., 1990***). YFP-*Hs*Importin-β and 3xGFP were used as positive and negative controls, respectively. To select only intact nuclei for our analyses, each reaction was supplemented with 155 kDa TRITC-labeled dextran, and only nuclei that excluded TRITC-dextran were analyzed (see ***Figure 5—figure supplement 1*** and materials and methods section for details).

As expected, YFP-importin-β efficiently translocated into nuclei independent of the presence of a nuclear transport mixture (*Xenopus* cytosol plus energy regeneration system). At the same time no significant nuclear translocation of 85 kDa 3xGFP was observed in any of the tested conditions, consistent with the approximately 40–60 kDa diffusion permeability cutoff of intact NPCs (***Figure 5A,B***) (***Mohr et al., 2009***). Strikingly, both YFP-*Mt*Nup188 and YFP-*Mt*Nup192 mimicked the behavior of YFP-*Hs*Importin-β and accumulated in intact nuclei even in the absence of a nuclear transport mixture. Nuclear uptake was completely blocked in the presence of WGA, a broad inhibitor of NPC-mediated transport (***Finlay et al., 1987***; ***Mohr et al., 2009***) (***Figure 5A,B***). Furthermore, translocation of both nucleoporins, and of YFP-*Hs*Importin-β, was strongly inhibited upon addition of an unlabeled importin-β (***Figure 5C,D***). Together these results demonstrate that the translocation of YFP-*Mt*Nup188 and YFP-*Mt*Nup192 occurs through NPCs. Since the sizes of YFP-*Mt*Nup188 and YFP-*Mt*Nup192 (~220 kDa) are far above the diffusion permeability cutoff of the NPCs this efficient nuclear translocation cannot be explained by simple passive diffusion.

To examine more directly if YFP-*Mt*Nup188 and YFP-*Mt*Nup192 cross the NPC by facilitated diffusion we measured the rates of their NPC translocation and compared them to the inert 3xGFP, which is almost threefold smaller compared to the model transport receptor YFP-*Hs*Importin-β. We followed the nuclear accumulation kinetics of each protein and calculated initial nuclear translocation rates by fitting each set of kinetic data to a single exponential decay function (***Figure 6***, see also

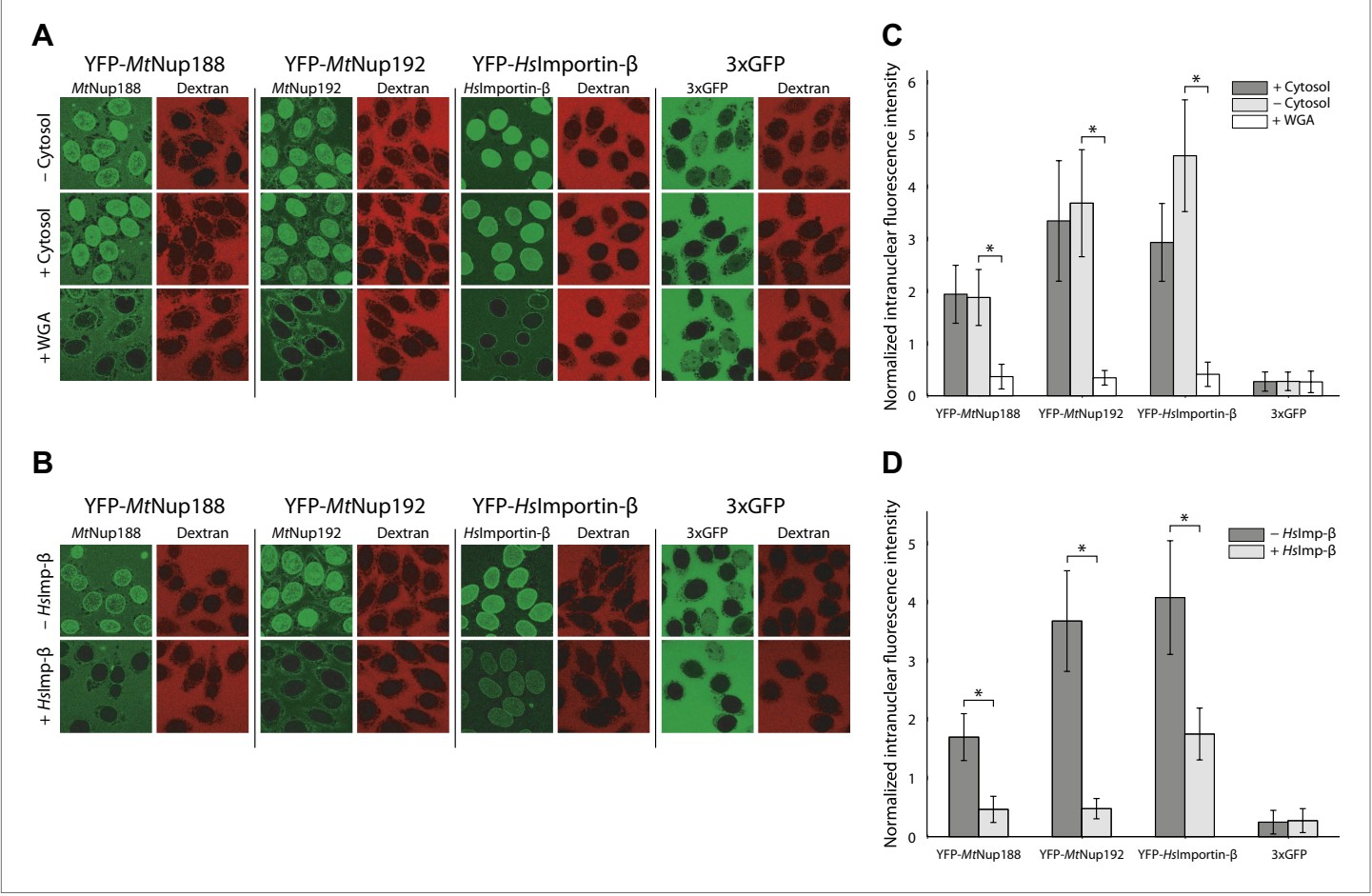

**Figure 5**. Nuclear translocation of Nup188 and Nup192 in digitonin-permeabilized HeLa cells. (**A** and **B**) The nuclear translocation of YFP-*Mt*Nup188 and YFP-*Mt*Nup192 (green) was monitored in digitonin-permeabilized HeLa cells in transport buffer alone (−cytosol), in transport buffer containing *Xenopus* cytosol and an energy regenerating mix (+cytosol), or in transport buffer after pre-treatment of nuclei with WGA (+WGA) (**A**). Similar assays were performed in transport buffer with or without the addition of unlabeled *Hs*Importin-β (**B**). 155kD TRITC-dextran (red) was used to check nuclear integrity. YFP-*Hs*Importin-β and 3xGFP were used as positive and negative controls for facilitated NPC translocation, respectively. (**C** and **D**) Quantitative analysis of the protein translocation experiments described in (**A** and **B**), respectively. The intranuclear fluorescence intensities of the proteins were quantified and normalized against the background (extranuclear) fluorescence. Bar graphs represent mean values ± standard deviations. Asterisks (*) indicate a significant difference in median value using the Mann-Whitney U test (p<0.01; >50 nuclei for each condition).

The following figure supplements are available for figure 5:

**Figure supplement 1**. Outline of the nuclear translocation assay.

'Materials and methods' for details). While both nucleoporins translocated slower than YFP-*Hs*Importin-β, their translocation rates exceeded the values of 3xGFP by at least 30-fold indicating that both proteins indeed cross the NPCs in a facilitated manner (***Figure 6***). The digitonin permeabilization protocol does not remove all endogenous NTRs (***Nachury and Weis, 1999***). To test whether the nuclear translocation rates of *Mt*Nup188 or *Mt*Nup192 can be enhanced by the removal of endogenous NTRs that remain tightly associated with NPCs we performed similar analyses with nuclei that were pre-treated with RanGTP (***Nachury and Weis, 1999***). While we saw an increase in the YFP-*Mt*Nup188 translocation rate, no effect on YFP-*Mt*Nup192 was observed. Under these conditions the translocation kinetics of *Mt*Nup188 and *Mt*Nup192 were approximately equal, but still about fourfold below the values for YFP-*Hs*Importin-β (***Figure 6*** and ***Figure 6—figure supplement 1***). This suggests that the slower nuclear translocation rates of *Mt*Nup188 and *Mt*Nup192 are intrinsic properties of these two proteins. Interestingly, both *Mt*Nup188 and *Mt*Nup192, unlike *Hs*Importin-β accumulated within nuclei in

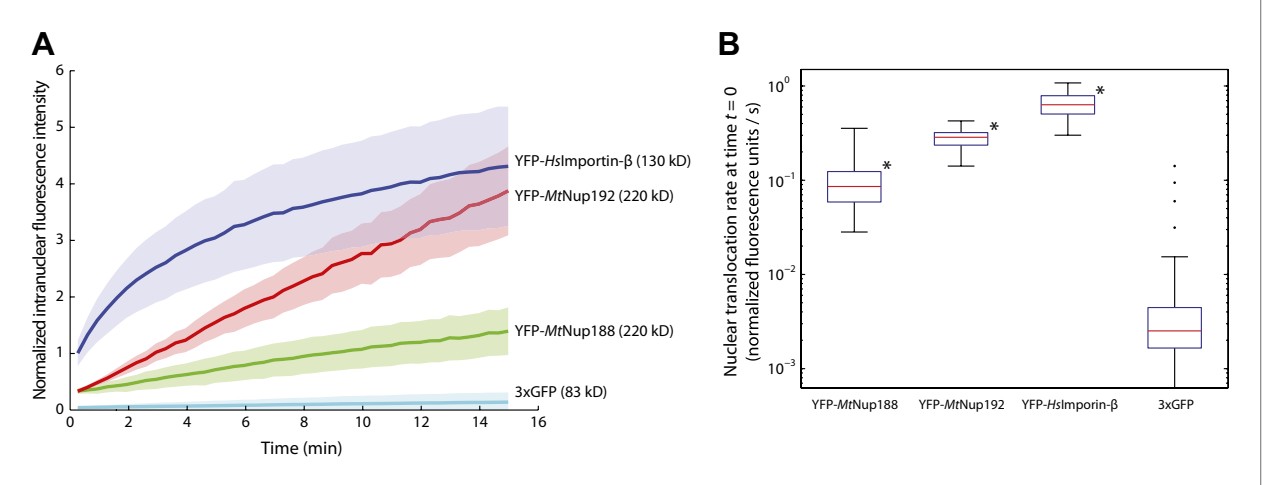

**Figure 6**. Kinetics of Nup188 and Nup192 nuclear translocation. (**A**) The nuclear accumulation kinetics of the scaffold nucleoporins in transport buffer as compared to YFP-*Hs*Importin-β and 3xGFP. Nuclear translocation reactions were imaged immediately after the addition of the fluorescent proteins and TRITC-dextran. Non-intact nuclei permeable to TRITC-dextran were excluded from the analysis. The graph represents mean values ± standard deviations. (**B**) Quantitative analysis of the initial nuclear translocation rates (equal to the total protein translocation rate through NPCs) for the experiments described in (**A**). The nuclear accumulation dataset for each nucleus (>30 for each condition) was separately fitted to a single-exponential curve, and the derivative at time $t = 0$ representing the initial nuclear translocation rate were computed. Asterisks (*) denote a significant difference in median value from the 3xGFP rate using the Mann-Whitney U test ($p < 0.01$). Box plots show the median, first and third quartiles, and non-outlier extrema. Values greater than six standard deviations from the mean are marked as outliers.

The following figure supplements are available for figure 6:

**Figure supplement 1**. Effect of RanGTP on the nuclear translocation kinetics of Nup188 and Nup192.

**Figure supplement 2**. Nuclear accumulation of Nup188 and Nup192 upon 10-fold dilution.

the absence of exogenous energy without reaching notable saturation during the time frame of our experiments (*Figure 6*). Furthermore, both proteins continued to accumulate in nuclei even after subsequent 10-fold dilution, suggesting that they have strong intranuclear binding sites (*Figure 6—figure supplement 2*). In summary, these results demonstrate that *Mt*Nup188 and *Mt*Nup192 share functional characteristics with NTRs and translocate across NPCs by facilitated diffusion without a requirement for other cytosolic factors.

## Discussion

The analysis of the largest universally conserved nucleoporin Nup188, which we present here, fills the last gap in the inventory of architectural nucleoporin structures. We find that the topology of Nup188 is distinctly different from all other scaffold nups, except for the related paralog Nup192, in that it appears to be inherently flexible. Negative stain EM analysis (*Figure 3C*) reveals that the stacked helical domain can adopt a 'closed' conformation seen in the crystal, but that the protein can also 'open' up. A similar degree of flexibility was recently also observed for Nup192 by negative stain EM analysis (*Flemming et al., 2012*). Other, mainly helical scaffold nucleoporins contain elements that prevent such flexibility. For instance, the ACE1 class of nucleoporins (Nic96, Nup84, Nup85, Nup145C) share a distinct foldback architecture where a string of helices latches onto the core helical stack, thereby prohibiting the bending of neighboring helical elements (*Brohawn et al., 2008*, *2009*). Three conserved scaffold nups (Nup120, Nup133, Nup170) share a topology where an N-terminal β-propeller is linked to a C-terminal helical stack domain. For each of these three nucleoporins there is structural evidence that stretches within the helical elements are somewhat flexible, but likely not to the extent that is observed for Nup188 and Nup192 (*Whittle and Schwartz, 2009*; *Bilokapic and Schwartz, 2012*). How does this exceptional flexibility shape the functional role of Nup188 and Nup192? Based on current data, we can only speculate. It is conceivable that structural changes within these large nups are critical during the assembly and biosynthesis of the NPC. Alternatively, plasticity of Nup188,

Nup192, and the Nic96 complex as a whole, may be important for the nuclear transport function of the NPC, for example, by accommodating large soluble cargos or regulating the translocation of inner nuclear membrane proteins.

Although Nup188 and Nup192 are paralogous proteins with very similar structural arrangements, they have acquired distinct functions during evolution. For instance, the depletion of Nup188 was shown to increase the size limit for the transport of inner nuclear membrane proteins, but this was not observed upon removal of Nup205 (the Nup192 homolog in vertebrates) (*Theerthagiri et al., 2010*). Our structure unexpectedly shows that Nup188 contains an SH3-like domain, but a similar domain cannot be detected in Nup192. Whether this SH3-like domain contributes to the functional diversification of these two large nups in transmembrane transport is currently unknown and is an interesting question for future studies. SH3 domains are generally involved in protein-protein interactions but the typical residues needed for the interaction with known SH3 substrates, are not conserved in Nup188. Therefore, it will be important to identify the binding partners of the SH3-like domain in Nup188 and to directly test whether this domain functions, for example, in the transport of transmembrane cargos.

So far, soluble NTRs and scaffold nucleoporins were considered to be functionally and evolutionarily separated groups of proteins that were already present in the last eukaryotic common ancestor each very likely originating from relatively few ancestor proteins through multiple gene duplication events (*Mans et al., 2004*). Intriguingly, our crystallographic analysis revealed that Nup188 is structurally related to NTRs. Moreover, single particle EM analysis showed that both Nup188 and its paralog, Nup192, resemble the overall shapes of NTRs and display exceptional flexibility characteristic for NTRs (*Figure 3*) (*Conti et al., 2006*). However, the structural similarity is rather general and not specific to a particular NTR. Following the entire 202 kDa structure, one finds a patchwork of 10–40 kDa substructures that are closer related to one NTR vs another. Based on the structure alone it is not possible to conclude a common phylogenetic origin; too many proteins involved in diverse protein–protein interactions adopt stacked helical domains and these domains likely have evolved multiple times throughout evolution (*Aravind et al., 2006*). We therefore tested whether Nup188 and Nup192 have functional similarities with NTRs. All known NTRs bind FG-repeats, typically exhibiting various binding sites dispersed over their surface (*Bednenko et al., 2003*). Intriguingly, both Nup188 and the related Nup192 directly and specifically bind to FG-repeats (*Figure 4A,B*). Furthermore, both Nup188 as well as Nup192 can translocate across intact NPCs in vitro, again recapitulating the behavior of NTRs (*Figure 5*). By contrast, we do not observe RanGTP binding (*Figure 4C,D*), another characteristic of the importin–β family of NTRs (*Floer and Blobel, 1996*; *Chook and Blobel, 1999*; *Cook et al., 2007*). Unlike the FG-binding pockets, the RanGTP interaction site is much larger and sequence-conserved, thus easier to recognize. Neither Nup188 nor Nup192 show surface conservation consistent with Ran binding (*Cook et al., 2007*). Yet, the combination of shared FG-binding activity, common NPC translocation properties combined and general structural similarity make it tempting to speculate that Nup188, Nup192 and NTRs, indeed, might have a common evolutionary origin. This blurs the line of a strict functional separation between nucleoporins and NTRs, and allows for a number of interesting hypotheses. Could it be that the ancestral NPC had only a few core nucleoporins, from which NTRs evolved to become the separate, 'soluble phase'? Or alternatively, could some soluble NTRs or shuttling cargos have evolved to bind tightly to the original NPC membrane coat and become part of the architectural scaffold?

The finding that architectural nucleoporins bind FG-repeats and display NTR-like properties, raises the question of the biological function of these properties. Once incorporated into NPCs Nup188 and Nup192 are unlikely to shuttle across the nuclear envelope. The very low mobility of their common binding partner, Nup93 in vertebrate NPCs and an exceptionally slow protein turnover suggest that both proteins are very stable NPC components (*Rabut et al., 2004*; *Savas et al., 2012*). However, the NTR-like activity of Nup188 and Nup192 could be critical for de novo NPC biosynthesis during interphase nuclear growth in higher eukaryotes or in organisms, which undergo a closed mitosis when new NPCs assemble de novo into the intact nuclear envelope (*D'Angelo et al., 2006*; *Dultz and Ellenberg, 2010*; *Menendez-Benito et al., 2013*). Interphase NPC assembly seems to rely on steps that occur on both the cytoplasmic and the nucleoplasmic side of the nuclear envelope and thus nuclear translocation of NPC components through already existing pores might be critical for new NPC assembly in intact nuclei. Another possibility is that FG-scaffold interactions are important for the structural integrity of the NPC by enabling connections between subcomplexes. Interactions within specific subcomplexes have been studied extensively and are rather well characterized. In contrast,

the interaction between subcomplexes is much more poorly understood, and it remains unclear how subcomplexes come together to assemble into the oligomeric NPC ring structure. Interestingly, the ability to interact with FG-repeats may extend to other scaffold nucleoporins as well. Nic96 and its vertebrate ortholog, Nup93 bind GLFG-type domains in vitro (*Schrader et al., 2008*; *Xu and Powers, 2013*). Furthermore, budding yeast scaffold nucleoporins can be biochemically purified together with NTRs from yeast extracts using FG-repeat regions as a bait (*Allen et al., 2001*). Therefore, the interactions between FG-repeats and scaffold nups could possibly be part of the glue that holds the NPC scaffold together.

Finally, the FG-binding properties that we have discovered here might also play a role in the correct establishment of the NPC selectivity barrier for soluble or transmembrane cargos by organizing the FG meshwork within the NPC channel. FG domain interactions with scaffold Nups could connect FG-repeat regions to the wall of the NPC questioning models in which FG-repeats function exclusively within the central channel of the NPC to mediate nuclear transport. Future experiments will be needed to test these models and to characterize the role of the FG-scaffold interactions in NPC biogenesis, structure or function.

## Materials and methods

### Construct generation

Expression constructs were PCR amplified from genomic DNA of the thermophillic fungus *Myceliophthora thermophila* (*Mt*). Two introns were removed using inverse PCR. Nup188 (residue 1–1827), Nup188N (residue 1–1160) and Nup188C (residue 1445–1827) were cloned into a pETDuet-1 vector from EMD Millipore (Billerica, MA) and N-terminally fused with a 3C protease cleavable 10xHis-7xArg-SUMO tag. The full ORF of *S. cerevisiae* (*Sc*) Nup188 was PCR amplified and inserted into pETDuet-1. It was fused to an N-terminal 3C cleavable 10xHis-7xArg-SUMO tag and a C-terminal 6xHis tag. Full-length *Mt*Nup192 was amplified from genomic DNA and its three introns removed by inverse PCR. Nup192 (residue 1–1734), Nup192 (residue 1–570) and Nup192 (residue 1–1034) were cloned into pETDuet-1, with an N-terminal 3C cleavable 10xHis-7xArg-SUMO tag and a C-terminal 3C cleavable 9xArg-7xHis tag. YFP-*Mt*Nup188 (residue 1–1827) and YFP-*Mt*Nup192 (residue 1–1734) expression constructs were derived from the respective non-labeled expression constructs (see above), by in-frame insertion of the YFP coding sequence after the respective N-terminal affinity tags. The *Sc*Kap95 expression plasmid was constructed by insertion of PCR-amplified *Sc*Kap95 ORF into pET30-a(+) His-tag expression vector. The bacterial expression plasmids for GST-*Sc*NSP1(1–603), GST-*Sc*Nup100(1–307) and were made by insertion of PCR-amplified coding sequences of the respective FG-repeat segments into the pGEX-2T vector.

### Protein production

All *Mt* proteins were expressed at 18°C in *Escherichia coli* BL21(DE3) RIL cells and first purified on Ni-affinity resin. When used for structure determination the eluted proteins were loaded on a HiTrapS FF column from GE Healthcare (Piscataway, NJ) and eluted with a linear NaCl gradient. Affinity tags were removed with 3C protease, followed by a second HiTrapS FF column purification step. Finally, the proteins were purified on a Superdex S200 10/300 column from GE Healthcare (Piscataway, NJ) in gel filtration buffer (10 mM HEPES pH 7.4, 150 mM NaCl and 1 mM DTT). Selenomethionine-derivatized protein was produced as described (*Brohawn et al., 2008*) and purified identical to the native protein, except that reducing agent was kept at 5 mM in all buffers. For FG-pulldowns, Ran-binding assays and protein translocation assays, the *Mt* proteins were purified using the same protocol that was used for structure determination, except for omitting the first HiTrapS FF column step. *Hs*Importin-β versions and *Sc*Kap95 were expressed and purified on Ni-affinity resin, followed by dialysis against the gel filtration buffer. ZZ-*Hs*RanQ69L was produced essentially as described in (*Nachury et al., 2001*). Purification of human Ran and nucleotide loading was performed as described in (*Lowe et al., 2010*). All GST-fusions of budding yeast FG-repeat regions were expressed in BL21(DE3) cells and purified on GSH-beads from GE Healthcare (Piscataway, NJ) in PBS pH 7.4 supplemented with 10 mM DTT, 0.1 mM PMSF and 0.5% Triton X-100. The beads were washed with PBS and proteins were eluted with glutathione and dialyzed against gel filtration buffer.

### Crystallization

Small needles of Nup188N grew in hanging drops containing 1.5 µl of protein at 4–6 mg/ml and 1.5 µl of precipitant (0.1 M MES pH 6.5, 4.5–7% (w/v) PEG 4000, 150 mM ammonium sulfate and 1 mM DTT)

at 16°C. Large (300 × 50 × 50 µm) diffraction quality crystals were obtained after micro-seeding and the addition of 1–2.5% (v/v) tert-butanol. Crystals were cryoprotected by briefly soaking in precipitant supplemented with 20% (v/v) PEG 200 and were flash-frozen in liquid nitrogen. Crystal of Nup188C grew in hanging drops containing 1.5 µl of protein at 3–5 mg/ml and 1.5 µl of precipitant (18–23% (w/v) PEG 3350, 150 mM tri-ammonium citrate and 1 mM DTT) at 16°C. Crystals were cryoprotected by dialysis into precipitant with 35% (w/v) PEG 3350 and flash-frozen in liquid nitrogen.

## Data collection and structure determination

Data were collected at the NE-CAT beamlines 24ID-C at Argonne National Laboratory. HKL2000 (*Otwinowski and Minor, 1997*) was used to reduce data. The structure of Nup188N was solved using SAD phases (usable to 3 Å) from a selenomethionine-substituted crystal. Out of 23 possible sites SHELXC/D/E (*Sheldrick, 2010*) found 22 selenium sites, which were refined in Phaser-EP (*McCoy et al., 2007*). An interpretable map was obtained after density modification using Parrot in the CCP4 suite (*Winn et al., 2011*). After most of the model was built, native data was used for the final rounds of model building and refinement using Coot (*Emsley et al., 2010*) and Phenix (*Adams et al., 2010*). Five loops (residue 85–91, 447–459, 673–682, 886–902, 963–970) and the very C-terminal 11 residues could not be built, due to poorly defined electron density. The structure of Nup188C was also solved using SAD phases from a selenomethionine-substituted crystal. All three possible sites of both molecules in the asymmetric unit were found using *phenix.autosol*. In the initial solvent-flattened electron density map several helices of both Nup188C molecules were well defined and could be built. In general, one copy of Nup188C is better defined then the other, presumably because of tighter crystal packing. Phases were improved by combining SAD phases with model phases using Phaser-EP. Even though the overall structure is similar, both Nup188C molecules were built independently, without using non-crystallographic symmetry restraints. The conformation between both molecules is different enough, that NCS was not beneficial even in the early rounds of refinement. The structure is complete except for three loops (residues 1495–1509, 1534–1538, 1705–1720) and the very C-terminal four residues, all of which are poorly defined in the electron density map. Structure figures were made using PyMol (Schrödinger LLC) and structural superposition carried out using the Dali server (*Holm and Rosenström, 2010*) and Coot.

## Single-particle EM data collection and analysis

Recombinantly expressed and purified full-length *Mt*Nup188, *Sc*Nup188 and *Mt*Nup192 at ~1.0 mg/ml were diluted with the gel filtration buffer (10 mM HEPES pH 7.4, 150 mM NaCl, 1 mM DTT) to 5 µg/ml and negatively stained with 2% uranyl acetate on continuous carbon-film grids. Single-particle CCD images of these three specimens were recorded on an FEI Tecnai Spirit electron microscope at 80 keV, 1–2 µm defocus and 60,000x magnification with a pixel size of 3.6 Å. A total of 1252 particles of *Mt*Nup188, 3000 particles of *Sc*Nup188, and 1408 particles of *Mt*Nup192 were boxed into separate stacks and were then each subjected to 2D classification using the single-particle data analysis package PARTICLE (www.sbgrid.org/software/title/PARTICLE). The 'direct particle classification' method implemented in PARTICLE does not require image pre-alignment, therefore the result is objective and free of any alignment error or reference bias.

## Pulldown assays

All FG-pulldown assays were performed in pulldown buffer (10 mM HEPES pH 7.5, 160 mM KOAc, 1 mM MgOAc2, 1 mM DTT, 0.01% Triton X-100). Bound proteins were eluted with 1 M NaCl elution buffer (10 mM Tris-HCl pH 8.0, 1 M NaCl, 0.1% Tween-20, 5 mM BME), followed by SDS-sample buffer. All the procedures except otherwise mentioned were performed at room temperature. The GST-FG fusions were pre-bound to GSH beads in the following amounts per pulldown reaction: 25 µl GSH resin; 10 nmol GST-Nup116(358–458); 10 nmol GST-Nup116(358–458) 10xFA; 5.0 nmol GST-Nup100(1–307); 2.5 nmol GST-Nup100(1–610); 3.0 nmol GST-NSP1(1–603). The beads were washed twice with the elution buffer re-equilibrated with pulldown buffer and aliquoted into separate siliconized tubes. The pulldown input proteins were pre-mixed with bacterial extract proteins in pulldown buffer to yield the following amounts per reaction: 0.5 ml pulldown buffer containing either *Mt*Nup188 (100 pmol), *Mt*Nup192 (100 pmol), *Sc*Kap95 (200 pmol) or 3xGFP (500 pmol) plus 200 µg bacterial extract proteins. The pulldown reactions were gently agitated for 20 min, quickly washed twice with 1 ml ice-cold pulldown buffer and bound proteins were eluted by 100 µl of the elution buffer followed by 100 µl of SDS-sample buffer. Salt eluates (1M NaCl) were precipitated with methanol-chloroform

and re-solubilized in 50 µl SDS-sample buffer. The pulldown input samples and their components were prepared for SDS-PAGE essentially as described for the salt eluates. The gels were stained with SYPRO Ruby (Invitrogen) and imaged with a UV-transilluminator equipped with a CCD-camera. The digital images were analyzed and prepared using ImageJ software.

## Ran binding assays

ZZ-*Hs*RanQ69L was pre-mixed with either Ran binding buffer alone (10 mM HEPES pH 7.4, 150 mM NaCl, 2 mM MgOAc2, 1 mM DTT), *Sc*Kap95, *Mt*Nup188FL or *Mt*Nup192FL to yield 2 µM concentration of each protein. The binding reactions were incubated on ice for 20 min, passed through 0.45 µm filter unit and subjected to gel filtration using Superdex 200 10/300 (GE Healthcare) in the Ran binding buffer. 0.5 ml elution fractions were collected, separated by SDS-PAGE and stained with SYPRO Ruby to visualize *Sc*Kap95, *Mt*Nup188FL or *Mt*Nup192FL. ZZ-*Hs*RanQ69L was visualized by Western Blotting using rabbit IgG followed by fluorophore conjugated secondary antibodies.

## Nuclear translocation assays

HeLa cells were cultured in DMEM media (+10% FBS) and plated on glass-bottomed poly-lysine coated chambers (MatTek, Ashland, MA) at a seeding concentration of $2.5 \times 10^5$ cells/ml the day prior to use. The cell permeabilization protocol is based on that of (*Adam et al., 1990*) Briefly, cells were washed with PBS (137 mM NaCl, 2.7 mM KCl, 8 mM $Na_2HPO_4$, 2 mM $KH_2PO_4$, pH 7.4) followed by a wash with permeabilization buffer (50 mM HEPES pH7.3, 50 mM KOAc, 8 mM $MgCl_2$). The cells were then treated with 50 µg/ml digitonin in permeabilization buffer for 5 min followed by three washes with transport buffer (20 mM HEPES pH 7.3, 110 mM KOAc, 5 mM NaOAc, 2 mM MgOAc). After the final wash, the buffer was removed and 100 µl of transport buffer containing the experimental protein mixtures was added to the cells (see *Figure 5—figure supplement 1*). Concentrations used were: 1 µM YFP-*Mt*Nup188 /192, 1 µM YFP- *Hs*Importin -β, 1 µM *Hs*Importin-β, 1 µM 3xGFP, 200 µg/mL 155 kDa TRITC-dextran (Sigma-Aldrich, St. Louis, MO), 2 mM DTT, 100 µg/mL WGA (Sigma-Aldrich), and 5% v/v cytosolic extract. Cytosolic extract from *Xenopus laevis* oocytes was prepared as described in (*Levy and Heald, 2010*). Experiments using the cytosolic extract were supplemented with an energy regenerating system consisting of 2 mM GTP (Roche, Indianapolis, IN), 100 µM ATP (Roche), 4 mM creatine phosphate (Roche), and 20 U/ml creatine kinase (units of specific activity given by Roche). For WGA pre-treatment, 100 µg/mL WGA was first incubated with the nuclei for 10 min, removed, and then the experimental mix was added to the nuclei. For experiments involving a Ran wash, 5 µM RanGDP and the energy regenerating system were added to the nuclei for 10 min followed by three washes with transport buffer, and then the experimental mix was added to the nuclei.

## Confocal imaging

After the transport reaction was allowed to run for 15 min at room temperature, images were acquired on a Zeiss LSM 700 laser scanning confocal microscope (63× oil objective; EYFP, EGFP, & TRITC channels; 1 airy unit pinhole; 4 line averaging) using the Zen 2011 imaging software package (Carl Zeiss, Inc.). For the time-lapse experiments the confocal imaging was started at the same time as the transport mix was added to the cells (63× oil objective; EYFP, EGFP, & TRITC channels; 1 airy unit pinhole; no line averaging; 15 min run time; 20 s intervals).

## Image and data analysis

Fluorescence quantification and data analysis were performed using custom written code in MATLAB (The MathWorks, Inc.). The fluorescence intensity inside each nucleus and the average background fluorescence intensity were determined for both the YFP/GFP and TRITC channels. The intranuclear to extranuclear ratios were then calculated giving normalized fluorescence intensities for both the protein of interest and the dextran. All nuclei with normalized dextran fluorescence ratio greater than 0.3 were considered to be 'non-intact' (permeable to the dextran) and rejected. Only 'intact' nuclei were subject to further analysis (see *Figure 5—figure supplement 1*). For time-lapse experiments, the time-dependent normalized fluorescence intensity of the protein of interest was determined for each nucleus and then fit to a single exponential curve, $y(t) = A*(1-\exp[k*t])+C$, where $A$, $k$, and $C$ are fit parameters, and $t$ is time. The initial rate of nuclear accumulation is given by the time derivative at $t = 0$ (i.e., $-Ak$).

## Acknowledgements

We would like to thank Michael Rexach for providing expression plasmids for GST-Nup116(348–458) and GST-Nup116(348–458)-10xFA. This work is based upon research conducted at the Advanced Photon Source on the Northeastern Collaborative Access Team beamlines, which are supported by award RR-15301 from the National Center for Research Resources at the NIH.

## Additional information

### Competing interests

KW: Reviewing editor, *eLife*. The other authors declare that no competing interests exist.

### Funding

| Funder | Grant reference number | Author |
| --- | --- | --- |
| National Institutes of Health | R01GM077537 | Thomas U Schwartz |
| National Institutes of Health | R01GM058065 | Karsten Weis |
| Lundbeck Foundation | | Kasper R Andersen |
| DFF Sapere Aude | | Kasper R Andersen |
| National Cancer Institute | U54CA143836 | Jeffrey H Tang |

The funders had no role in study design, data collection and interpretation, or the decision to submit the work for publication.

### Author contributions

KRA, EO, JHT, Conception and design, Acquisition of data, Analysis and interpretation of data, Drafting or revising the article; PK, JZC, Acquisition of data, Analysis and interpretation of data; AU, Contributed unpublished essential data or reagents; JTL, Analysis and interpretation of data; KW, TUS, Conception and design, Analysis and interpretation of data, Drafting or revising the article

## Additional files

### Supplementary files

• Supplementary file 1. List of constructs used in this study.

### Major datasets

The following datasets were generated

| Author(s) | Year | Dataset title | Dataset ID and/or URL | Database, license, and accessibility information |
| --- | --- | --- | --- | --- |
| Schwartz TU, Andersen KR | 2013 | MtNup188N (1-1160) | 4KF7; http://www.rcsb.org/pdb/search/structidSearch.do?structureId=4KF7 | Publicly available at the RCSB Protein Data Bank (http://www.rcsb.org/) |
| Schwartz TU, Andersen KR | 2013 | MtNup188C (1445-1827) | 4KF8; http://www.rcsb.org/pdb/search/structidSearch.do?structureId=4KF8 | Publicly available at the RCSB Protein Data Bank (http://www.rcsb.org/) |

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
