## [Decision Letter]

Thank you for sending your work entitled “Scaffold nucleoporins Nup188 and Nup192 are structurally and functionally related to nuclear transport receptors” for consideration at *eLife*. Your article has been favorably evaluated by a Senior editor and 3 reviewers, one of whom is a member of our Board of Reviewing Editors.

The following individuals responsible for the peer review of your submission want to reveal their identity: Wolfram Antonin and Dirk Görlich (peer reviewers).

The Reviewing editor and the other reviewers discussed their comments before we reached this decision, and the Reviewing editor has assembled the following comments to help you prepare a revised submission.

Andersen and colleagues report on the structure of Nup188. Nup188 is a universally conserved scaffold nucleoporin having a related paralogue, Nup192. This structure is a very important step towards a long-term goal of the field, namely to obtain an atomic model of entire nuclear pore complexes (NPCs). Likewise, the finding of scaffold nups interacting with FG domains is of great mechanistic interest. We strongly recommend publication of this study in *eLife*. Before acceptance, however, the following points need to be addressed:

1) There are several instances of imprecise citations. For example, [62] should already be cited for their Nic96p structure and not just for Nic96p binding the Nsp1p complex.

2) The authors make a major point of Nup188 and Nup192 interacting with FG repeats. They give the impression as if this would be the first described instance of such interaction. This is, however, not the case. In fact, the already mentioned Schrader 2008 study makes the same point for Nic96p (which also is an all-alpha-helical structure). This should be acknowledged by an appropriate citation within the Introduction.

3) The title suggests a functional relation of Nups 188 and 192 to nuclear transport receptors. This is misleading. They share an interaction, but not a function: transport receptors carry cargo through NPCs, while Nups 188 and 192 make up the stationary phase. Please re-phrase.

4) The observation that the interaction between MtNup188 and FG repeats is more salt stable than the other FG interactions does not imply that this pair shows the tightest binding. An alternative explanation is that this binding relies more strongly on hydrophobic than on ionic interactions. Please re-phrase.

5) WGA is quoted as a specific NPC translocation inhibitor (Finlay, 1987). A later study (Mohr, 2009), however, demonstrated that WGA not only inhibits facilitated translocation, but also passive leakage through NPCs. Please re-phrase. The inhibition of nuclear entry of Nup188 and Nup192 by WGA can also not be taken as direct evidence for facilitated NPC passage of these Nups, but only as a proof that transport occurred through nuclear pores.

6) The authors show that importin beta inhibits NPC passage of Nup188 and Nup192, and explain the effect by competition of the two species for FG binding sites. This is the simplest but not the most likely explanation. Given that an NPC contains >5000 FG motifs, but can probably not accommodate more than 100 importin molecules containing ≈ 10 FG-binding sites each, it is clear that FG motifs per se will not become limiting. More likely explanations are volume exclusion effects and the introduction of additional meshes into the permeability barrier, caused by the multivalent FG-importin interactions. In line with that, it had been previously demonstrated that saturation with importins also lowers the passive exclusion limit of authentic NPCs (Mohr, 2009) or of pure FG domain systems (Frey, 2009; Jovanovic-Talisman, 2009). Please re-phrase.

7) The authors write that their findings question the current paradigm for the organization of FG repeats within the central NPC channel. Our feeling is that the new data only question M. Rout’s and R. Lim’s view of FG domains being entropically repelled from their anchor sites. They are, however, fully consistent with the previous discussion of Schrader (2008) and models that assume that FG domains form a barrier by the formation of physical meshes. It is now very nice to see that the scaffold of the NPC contributes directly to such mesh formation.

---

## [Author Response]

*1) There are several instances of imprecise citations. For example, [62] should already be cited for their Nic96p structure and not just for Nic96p binding the Nsp1p complex*.

We have carefully proofread the manuscript and have added additional references where necessary, including the reference to [62], for the Nic96 structure.

*2) The authors make a major point of Nup188 and Nup192 interacting with FG repeats. They give the impression as if this would be the first described instance of such interaction. This is, however, not the case. In fact, the already mentioned Schrader 2008 study makes the same point for Nic96p (which also is an all-alpha-helical structure). This should be acknowledged by an appropriate citation within the Introduction*.

We have corrected this and have added the sentence “Intriguingly, Nic96 and its vertebrate ortholog Nup93 were shown to bind FG-repeats in vitro”, and we have included the appropriate references to acknowledge that FG-binding was previously observed for Nic96.

*3) The title suggests a functional relation of Nups 188 and 192 to nuclear transport receptors. This is misleading. They share an interaction, but not a function: transport receptors carry cargo through NPCs, while Nups 188 and 192 make up the stationary phase. Please re-phrase*.

To avoid any confusion we have changed the title to “Scaffold nucleoporins Nup188 and Nup192 share structural and functional properties with nuclear transport receptors”.

*4) The observation that the interaction between MtNup188 and FG repeats is more salt stable than the other FG interactions does not imply that this pair shows the tightest binding. An alternative explanation is that this binding relies more strongly on hydrophobic than on ionic interactions. Please re-phrase*.

We rephrased the sentence in question and now write: “The only notable deviation from the FG-binding pattern was observed with *Mt*Nup188, which was relatively inefficiently eluted with 1M NaCl from Nup100(1-310) and Nup100(1-610), suggesting that *Mt*Nup188 either interacts with these FG-repeat regions more tightly than *Sc*Kap95 or that these interactions are more hydrophobic in nature (Figure 4—figure supplement 1).”

*5) WGA is quoted as a specific NPC translocation inhibitor (Finlay, 1987). A later study (Mohr, 2009), however, demonstrated that WGA not only inhibits facilitated translocation, but also passive leakage through NPCs. Please re-phrase. The inhibition of nuclear entry of Nup188 and Nup192 by WGA can also not be taken as direct evidence for facilitated NPC passage of these Nups, but only as a proof that transport occurred through nuclear pores*.

We concur and we have changed the description of these results accordingly. For details, see our response to point 6, below.

*6) The authors show that importin beta inhibits NPC passage of Nup188 and Nup192, and explain the effect by competition of the two species for FG binding sites. This is the simplest but not the most likely explanation. Given that an NPC contains >5000 FG motifs, but can probably not accommodate more than 100 importin molecules containing ≈ 10 FG-binding sites each, it is clear that FG motifs per se will not become limiting. More likely explanations are volume exclusion effects and the introduction of additional meshes into the permeability barrier, caused by the multivalent FG-importin interactions. In line with that, it had been previously demonstrated that saturation with importins also lowers the passive exclusion limit of authentic NPCs (Mohr, 2009) or of pure FG domain systems (Frey, 2009; Jovanovic-Talisman, 2009). Please re-phrase*.

We agree that we cannot exclude that the importin beta inhibition results can be attributed to volume exclusion. To address points 5 and 6, we have reworded the text, and now only conclude (a) that the inhibition by WGA and importin beta demonstrates that MtNup188 and MtNup192 translocate through the NPC, and (b) that MtNup188 and MtNup192 are too large to translocate through the NPC by simple passive diffusion.

*7) The authors write that their findings question the current paradigm for the organization of FG repeats within the central NPC channel. Our feeling is that the new data only question M. Rout’s and R. Lim’s view of FG domains being entropically repelled from their anchor sites. They are, however, fully consistent with the previous discussion of Schrader (2008) and models that assume that FG domains form a barrier by the formation of physical meshes. It is now very nice to see that the scaffold of the NPC contributes directly to such mesh formation*.

To avoid any misrepresentation of the current literature we have rephrased this sentence and now write: “FG domain interactions with scaffold Nups could connect FG-repeat regions to the wall of the NPC questioning models in which FG-repeats function exclusively within the central channel of the NPC to mediate nuclear transport.”